# Creative Territory and Gastronomy: Cultural, Economic, and Political Dimensions of Tourism in Historic Brazilian Cities

Alice Leoti [1], Francisco Antonio dos Anjos [2,*] and Raphaella Costa [3]

1 Department of Tourism, Jaguarão Campus, Universidade Federal do Pampa, Bagé 96015-000, Brazil; alicesilva@unipampa.edu.br

2 Postgraduate Degree in Tourism and Hospitality Management, Universidade do Vale do Itajaí, Itajaí 88220-000, Brazil

3 Costa Sur Turismo, Betim 32604-492, Brazil; raphaellacosta.consultoria@gmail.com

* Correspondence: anjos@univali.br; Tel.: +55-31-9172-7242

**Abstract:** This study aims to present the interrelation between gastronomy and creative territories, from the cultural, economic, and political dimensions of sustainable cultural tourism. This study used tourist spaces as creative territories in Brazilian historic cities. The methodological process of understanding the analytical dimensions was based on the regressive-progressive method of Henri Lefebvre that establishes three dialectical movements: the descriptive, the analytic-regressive, and the historical-genetic. This paper deals with the results obtained in the first of three movements: the descriptive process, which aims to know the object that is being studied; making use of public information; and allowing a generalized theorization. The gastronomic heritage is understood as an important experience; however, it still does not play a prominent role in the official identity of the historic towns. The results show that gastronomy, an intangible cultural heritage, is politically forgotten in the historic towns. A diversified range of cultural attractions turned to their built cultural patrimony was recognized, but only three, Pelotas, Sobral, and São João del Rei, have their cultural identity associated with local gastronomy. The tourist activities of the cities analyzed, in their majority, are focused on nature, which propitiates the development of sun and beach tourism, relegating cultural and creative tourism to the background. Culturally the historic cities have their cultural identities associated with the built cultural patrimony, and only Pelotas and Sobral develop actions directed to their gastronomic patrimony. The tourism economy in the historic cities is directly related to cultural and gastronomic activities. Politically, the cultural area seems to be better structured and managed in comparison with the tourism area.

**Keywords:** Brazilian historic cities; built cultural heritage; intangible heritage; gastronomy; creative territory; cultural tourism

## 1. Introduction

The articulation between creative territory and gastronomy has a vital role in the tourist experience, being able to provide the tourist an approximation with the cultural habits of the local community in which they visit. Creative tourism is an alternative form in relation to mass tourism, since it aims to break the paradigm of mass travel, where residents cease to be mere service providers and become the authors and producers of culture and local heritage [1].

The factors that motivated this research are associated with the absence of research on the interaction between built heritage and intangible heritage, in this case studied through gastronomy, and how this relationship produced a creative territory. To carry out this study, the premise was the existence of a significant amount of built heritage; in this sense, it was chosen to research historic cities. From these definitions, other criteria were established, such as the size of the city and its regional attractiveness for cultural motivations.

In this sense, this study aims to present the interrelation between gastronomy and creative territories, from the cultural, economic, and political dimensions of sustainable cultural tourism. This study used tourist spaces as creative territories in Brazilian historical cities that have been recognized as Urban Set by the Institute of National Historical and Artistic Heritage (IPHAN) and that are configured as regional centers and are located in municipalities recognized as tourism in strata A or B of the Brazilian Tourism Map. On these criteria, eleven historic cities were identified: Cabo Frio/Rio de Janeiro, Nova Friburgo/Rio de Janeiro, Petrópolis/Rio de Janeiro, Corumbá/Mato Grosso do Sul, Porto Seguro/Bahia, Sobral/Ceará, Paranaguá/Paraná, Angra dos Reis/Rio de Janeiro, Parnaíba/Piauí, São João del Rei/Minas Gerais, and Pelotas/Rio Grande do Sul (Figure 1).

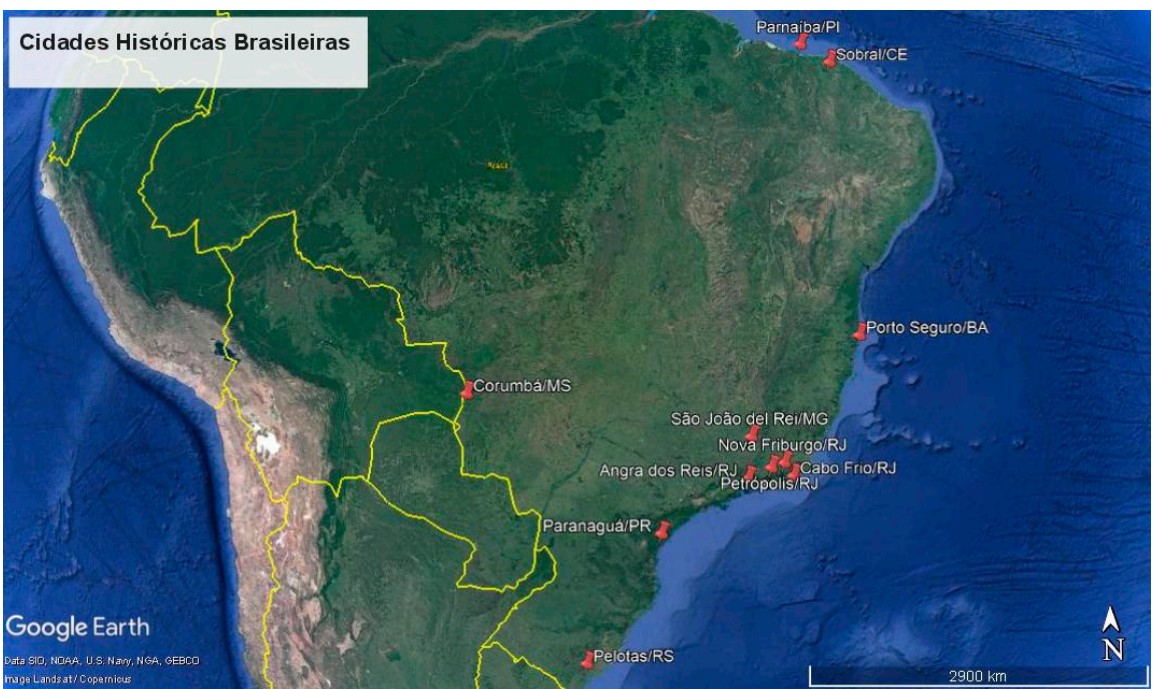

**Figure 1.** Location with the Brazilian historic cities under study. Source: Google Earth, 2022.

Understanding that a society can be analyzed through the social relations built from its multiple dimensions (economic, political, cultural, and geographic, among others), the research on Brazilian historic cities was structured from three dimensions: economic, cultural, and political [2].

This text is structured as follows: Introduction, which presents the factors that preceded this research, the cities studied, and how the study was structured; Literature review, which discusses the theoretical categories that formed the basis of this study, such as creative territory, identity, cultural heritage (the material studied being the built assets, and the immaterial analyzed from the gastronomy), and territory; Materials and methods, in which is presented the epistemological and methodological construction and an analysis model from the regressive-progressive method; Results, which demonstrates the findings of this research in the cultural, economic, and political dimensions; Conclusions, which presents the final considerations about the creative territory of the historic towns from the cultural, economic, and political dimensions; References, which contains the list of sources consulted for the elaboration of this study.

## 2. Literature Review

This section will discuss the themes that permeate tourism and its relationship with gastronomy in the Brazilian historic cities under study. In this sense, it is necessary to understand how the interaction between the built cultural heritage and the intangible

gastronomic heritage occurs and to what extent they can configure a creative territory in the historic centers.

In the field of tourism, the concept of territory as possessing historically built socio-spatial characteristics as a factor of socioeconomic development has emerged, thus demanding the use of talents and the elaboration of creative activities [3]. The concept of creative economy—a set of economic activities that are based on intellectual and economic capital—gave rise to the idea of the creative city—a territory which manages to develop economically and socially through its cultural aspects; in 2004 UNESCO created the Network of Creative Cities, and recently the notion of creative territory has been developed.

In Brazil, the Ministry of Culture [4] defined creative territory as "neighborhoods, cities or regions that present creative cultural potentials capable of promoting integral and sustainable development, combining preservation and promotion of their cultural and environmental values".

In this context, as well as in the creative economy, creative territories can be represented by five fields considered as productive and related to cultural tourism: (a) Field of Heritage: material heritage, immaterial heritage, archives, and museums; (b) Field of Cultural Expressions: handicrafts, popular cultures, indigenous cultures, Afro-Brazilian cultures, visual arts, and digital art; (c) Field of Performing Arts: dance, music, circus, and theater; (d) Field of Audiovisual/Book, Reading, and Literature: cinema and video, and publishing and printed media; (e) Field of Cultural and Functional Creations: fashion, design, and architecture [5]. Tourism has a strong relationship with the creative field of tangible and intangible cultural heritage—in this study, gastronomy—its social capital.

Discussing heritage presupposes talking about identities, because it can be defined as a symbolic synthesis of identity values, which help in the sense of belonging and the identification of a collectivity [6]. Prats [7] states that cultural heritage is "everything that is socially considered worthy of conservation regardless of its utilitarian interest. (...) Cultural heritage is an invention and a social construction". Cultural heritage has a utilitarian interest that lies in its ability to highlight and provide subsidies for understanding the identity of a society, as it is one of the elements that constitute a cultural identity. In parallel to the utilitarian and economic value, the symbolic value stands out as a reference and significance in the cultural order of a society [8]. It drives the shared culture and lived experiences.

The historic centers are the material record of social processes and historical events, which narrate the urban evolution, the territory occupation processes [9], and disputes and conflicts involved in it. The historic centers acquire the heritage sense by referring to different phases of the urbanization process of the cities.

In some cases, the historic centers went through emptying of functions, recreating uses and giving new meaning to the urban space [9]. In the process of territorial resignification from the tourism activity, one should start from a sustainable and creative development model, boosting the integration of the local community and its cultural assets [10,11]. In this process of resignification of historic centers, the boundaries between material cultural heritage and immaterial cultural heritage, as Andrade Jr. [12] notes, "have been diluted, and the very notion of architectural heritage can no longer be dissociated from the cultural practices it houses or to which it is associated".

Funari and Pelegrini [13] go along with the perspective pointed out [12] when talking about the inseparability of cultural practices linked to the PCM. Funari and Pelegrini [13] also point out that the PCM is predominantly associated with the elites, and that the PCI would be a counterpoint, since it values and favors the peripheral world.

Cultural practices encompass identity values and traditions and have symbolisms that characterize and individualize social groupings. These cultural practices are observed in behaviors, values, knowledge, and worldviews shared in a society. Pelegrini [14] affirms that "the healing, religious, and culinary knowledges are constituted as intangible heritage when they articulate the correlated experiences and experiences in the present and in the past". In this sense, we have Intangible Cultural Heritage when there is the interaction of

cultural practices between past and present but also when there is the feeling of continuity projecting into the future.

Gastronomy is recognized as PCI in Brazilian normative instruments, such as the Federal Constitution of 1988, Decree 3.551/2000, the National Program of Intangible Heritage (PNPI), and documents published by IPHAN and UNESCO, which demonstrate the relevance of safeguarding the gastronomic identity. "Food heritages are spaces of innovation (creativity) and permanence (traditions), where more and more value is placed not only on the moment of preparing food, but also on modes of socialization, cultural transmission and storytelling" [15].

Besides the legislative measures of protection to the Intangible Cultural Heritage, the dissemination of the tasting of typical dishes has been an important tool to safeguard gastronomic knowledge. Montanari [16] mentions that there are no trendy restaurants that do not boast a cuisine proposal linked to the territory of fresh food in the markets. Importantly, the search for the consolidation of a local gastronomic culture is founded in the past, in the search for roots, traditions, and history. Simultaneously, it is looking to the present—a moment of fragmentation and standardization of identity—and thinking of the future [15] by projecting a consolidation of regional and national identity with gastronomy as its basis [16].

Salvado, Ferreira, Serra & Marujo [15] state that the culinary arts are among the cultural elements that make up the intangible cultural elements of a national identity, since they are the result of traditional knowledge that to a certain extent narrates the historical and social evolution of a people. Therefore, this knowledge must be safeguarded and promoted.

Stefanutti [17] states that the word gastronomy has been understood in a broader sense, and that the term currently replaces other words, such as cuisine, cooking, and even food. Through the words of Stefanutti [17], it is observed that the use of the word gastronomy is not recent. The term was popularized by the work "The Physiology of Taste" (1995) by Brillat-Savarin, in which the idea that gastronomy is the act of eating and drinking appropriately was assimilated. The notion of gastronomy, in this sense, begins to understand dining as an opportunity to experience a culture through tourism.

The connection between tourism and gastronomy, created by Michelin and boosted by Ritz and Escoffier, has consolidated and gained space. Travelers seek new sensory experiences through gastronomy, which sharpens the five senses: sight, hearing, smell, touch, and taste. A multisensory experience provides the formation of affective gastronomic memories linked to a specific territory.

In this interaction between tourism and gastronomy, two elements stand out: intangible heritage and territory. The gastronomic culture of a society has in the combination of spices, climate and geography, preparation rituals and ways of serving, savoir-faire, and ways of living a set of values that mark the identity formation process of a people, forming part of the PCI. For Canaan [18], cultural heritage refers to the cultural and identity characteristics of a given territory, making it different from others. A trend of developing cultural attractions through events and festivals that add cultural values, such as gastronomy, can boost tourism beyond large urban centers [18,19].

The expression nicknamed by the French geographer Jean-Robert Pitte [16], "the geographical eating", denotes the idea of getting to know a culture through the typical cuisine of a particular region, and in it can be found aspects such as memories, social classes, rules, religions, and calendar foods. The obviousness of the relationship between gastronomic and territorial identity is naturalized because when an individual moves geographically, he already expects to find different meals from the ones he is used to.

In this context, the famous phrase "Tell me what you eat and I will tell you who you are" written by Jean Anthelme Brillat-Savarin in the work Physiology of Taste of 1826 [16] reveals in a historical perspective the personality and character of an individual through his taste and the way he eats. The way of eating denotes social belonging and locates one spatially. Food, in this perspective, is directly related to the resources available in the territory, which is an essential aspect of food culture.

The globalization process, from a food perspective, has a purpose of defining place as an inter-territorial, inter-regional, and international exchange, especially when talking about dishes that have historically been linked to territory. Globalization goes in the opposite direction of the uniqueness of the dishes, going towards the standardization, the mixing, the confusion, and the non-identification of the diverse cultures, that is, propitiating the cultural fusion of diverse cuisines.

At the same time that globalization promotes the erasing of territorial singularities, it also stimulates, paradoxically, the marking and reinforcement of regional and national identities as a way to differentiate from other territories and from other cultural identities. This paradoxical tendency contributes to the consolidation of national identities and regional units.

Tourism makes this access to culinary traditions and the cultural aspects involved in its production possible.

## 3. Materials and Methods

This research is structured based on the regressive-progressive method, proposed by Henri Lefebvre [20,21], in which three movements are performed in the research: the descriptive, the analytic-regressive, and the historical-genetic. The first movement of the method, the descriptive one, takes place through the observation of the object studied with information from experience and a general theory.

The second movement, called analytic-regressive, is the movement in which a decomposition of reality is made by dating social relations. Each social relation has an age and a date, and each element of material and immaterial culture also has its specific date.

The third movement, the historical-genetic or regressive-progressive, provides knowledge about the historical inconsistencies and about the temporal continuities and discontinuities that are part of the contradictions.

This text presents the results obtained by the procedures indicated in the first movement of the regressive-progressive method, which consists in reporting the detailed observation of the object and mapping the present. The descriptive movement takes place through the observation of the object studied with information from experience and a general theory. Several techniques can be used to help in the general description, such as observation, interviews, questionnaires, statistics, and others.

This research is of an applied nature, since it is based on the knowledge that will be produced about the tourism relations in the creative territories of the gastronomy field in historic cities. With regard to the objectives employed in the research, they can be understood as exploratory, explanatory, and descriptive.

In the first moment of the investigation, the procedures adopted were bibliographic and documental research. A bibliometric survey was carried out in the Integrated Library System of the Universidade do Vale do Itajaí (UNIVALI), which is integrated to several databases of national and international reach, with attempts at combinations of keywords that are presented in Table 1. The filters used were as follows: the keywords Boolean in the sentence, no full text, texts published in scientific journals in the period between 2014 and 2019.

Bibliographic surveys were also carried out in theses and dissertations banks and databases such as Taylor & Francis, Scielo, and Google Academic, among others. The bibliometric research was carried out with the purpose of delimiting the research problem and conceptualizing the following topics: built heritage, intangible heritage, gastronomy, territory, creative territory, and tourism.

Based on this epistemological and methodological approach used in this research, and with the purpose of contributing to future investigations, a model of relationships between the categories discussed was developed for the analysis of creative territory from the categories of identity, cultural heritage, and territory (Figure 2).

**Table 1.** Result of the bibliometric search performed until 10/10/2019 at UNIVALI's Integrated System.

| Keywords | Total Result | Analyzed Texts |
|---|---|---|
| tourism AND creative territory AND gastronomy OR cooking OR food | 57 | 18 |
| tourism AND creative territory AND food heritage | 21 | 2 |
| tourism (subject terms) AND creative territory AND Gastronomy OR gastronomy OR cuisine OR cooking (summary) | 16 | 4 |
| tourism AND creative territory AND food heritage (summary) | 20 | 0 |

Source: Elaborated by the Authors.

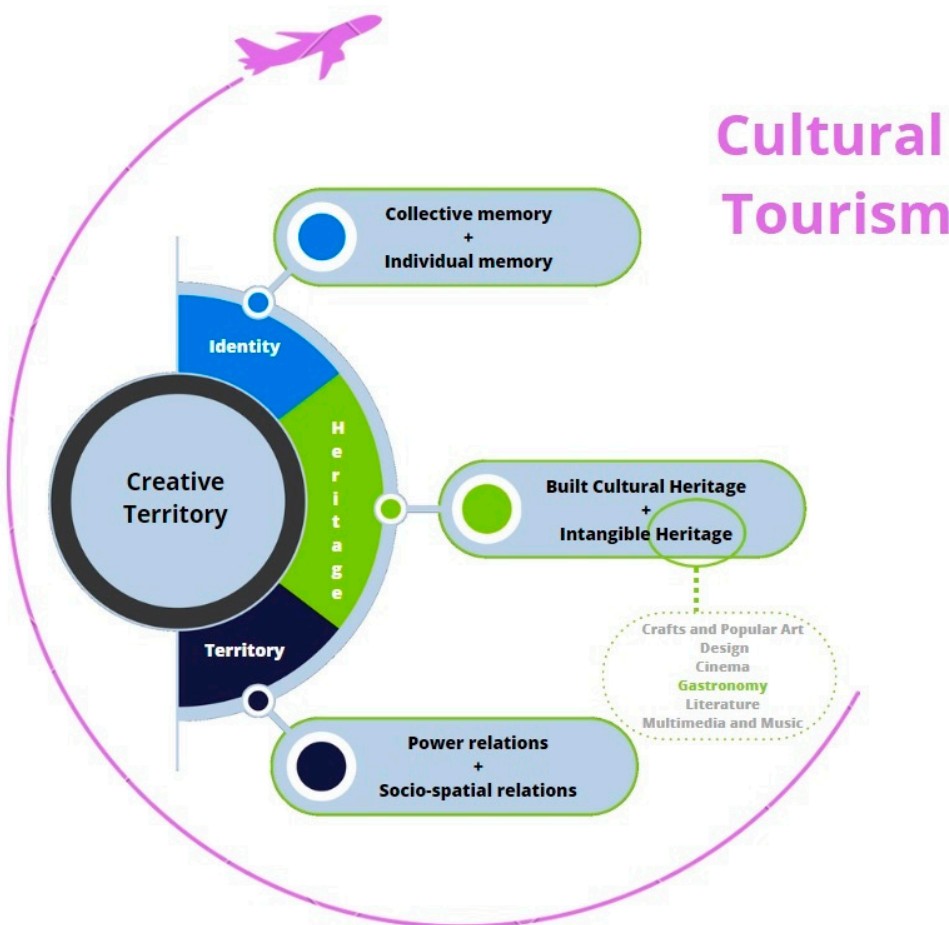

**Figure 2.** Research model of creative territory in historic cities. Source: Prepared by the authors.

The following criteria were established for the selection of the eleven Brazilian historic cities: to be a medium sized city (between 100,000 and 500,000 inhabitants); to have been categorized as A or B according to the Ministry of Tourism in the Brazilian Tourism Map; and the hierarchy established by the study of Regions of Influence of Cities (REGIC), which defines which municipalities are central in displacements for cultural purposes. This cutout recognized the cities of Sobral/CE, Paranaguá/PR, Parnaíba/PI, Pelotas/RS, São João del Rei/MG, Petrópolis/RJ, Angra dos Reis/RJ, Nova Friburgo/RJ, Porto Seguro/BA, Cabo Frio/RJ, and Corumbá/MS, located in eight Brazilian states in four geographical macroregions. The cities are analyzed from three dimensions: cultural, economic, and political.

The data collection for the documental research was carried out between May and June of 2021, in which the following were identified: tumbled material goods; employees in the

sectors of Food Services and Means of Accommodation; classification of the municipalities in the Brazilian Tourism Map; political structure in the sectors of culture and tourism; Food Services and Travel and Tourism Agencies; Cultural Displacement Centrality and Population Arrangements; immaterial goods; Means of Accommodation; and information about local handicraft, permanent events, and cultural attractions.

The collection of data on the number of food services and travel agencies in each city, in May and June 2021, was performed with Google Maps, a site widely used by tourists, making use of the Street View tool. This technique was also used to describe the shape of the built heritage.

For the collection of the number of lodging facilities, carried out in May and June 2021, the Booking website was used, which is an international website where citizens can book accommodations for vacations or travel. The site is now available in 43 languages. The search for accommodations included hotel, guesthouse, resort, farm hotel, bed and breakfast, historic hotel, hostel, and flat/apart-hotel. Private rentals of apartments or houses were excluded.

## 4. Results

The results will be presented in three spatial dimensions of the creative tourism territory: the Cultural dimension, the Economic dimension, and the Political dimension.

### 4.1. Cultural Dimension

In an analysis by the Cultural Dimension of tourism in the 11 historic cities that make up this study, it is verified that four cities, Sobral/CE (2000), Paranaguá/PR (2009), Parnaíba/PI (2017), and Pelotas/RS (2018), received the title of "Conjunto Urbano Tombado" in the 21st century. The only city to receive this title in the first half of the 20th century, precisely in 1938, was São João del Rei. The municipalities of São João del Rei/MG (1938), Petrópolis/RJ (1964), Angra dos Reis/RJ (1969), Nova Friburgo/RJ (1972), Porto Seguro/BA (1974), Cabo Frio/RJ (1989), and Corumbá (1993) were registered before IPHAN was created but only in 1994, when it was still the Serviço do Patrimônio Histórico e Artístico Nacional (SPHAN). When looking at the dating of the Urban Complexes, it is notable that most of them, six historic towns, were recognized for their historic value after the 1988 Federal Constitution.

The city that has the most cultural attractions is Petrópolis, with 73 attractions, followed by São João del Rei (41 attractions) and Pelotas (38 attractions). In these cities, it is observed that they have a cultural profile of their tourist attractions. Only Cabo Frio/RJ seems not to consider its history as a tourist attraction, in view of the fact that the City Hall website presents a report of the history of the city, but it has not identified any agency that offers visits to the historical sites. Despite apparently not considering its historical aspects as an attraction, Cabo Frio was considered by REGIC as one of the 30 main Brazilian municipalities with cultural centrality, counting 17 cultural attractions, a number that leaves it in the last position if compared with the other cities in this study. It is joined by four other historic cities: Pelotas, Porto Seguro, Parnaíba, and São João del Rei.

The hierarchy of cities created by the study of Regions of Influence of Cities (REGIC) demonstrates that the historic cities and their historic centers have assumed their cultural and patrimonial role throughout the process of urbanization of the cities [8]. They configure themselves as regional cultural centers [7], which, expresses the symbolic, utilitarian, and economic value that each historic city has, considering the region in which it is inserted. Just as the material and immaterial cultural heritage is an invention and social construction [7], the influence of cultural identity also assimilates this process.

All the historic cities studied have artisans' houses, artisans' associations, or events dedicated exclusively to artisanship. With regard to permanent events related to culture, Angra dos Reis stands out for having 11 events, and the second is São João del Rei, with eight events. This information can refer to a better integration between tourism and culture, but it is also possible to think that there is a greater structuring in the Political Dimension,

thus reflecting in the construction of an events calendar. It was not possible to identify the existence of events related to tourism and culture in Sobral/CE.

The existence of houses, associations, and events focused on handicrafts and culture corroborates the statement [10], who says that the relationship between the architectural cultural heritage and the intangible cultural heritage is increasingly inseparable since social practices, pointed as a counterpoint to the buildings, provide a sense of belonging and appreciation for the historic centers. Feelings so dear to the identity values and traditions, which relate to the cultural heritage, provide the encounter of the past with the present [14], resulting in unique experiences.

Pelotas stands out when it comes to the IPHAN listed PCM with 23 properties, thus forming the largest architectural collection among the historic cities under study. In terms of state-protected areas, the historic city in evidence is Petrópolis since it has 38 significant protected assets. When it comes to the PCI registered in IPHAN through the National Inventory of Cultural References (INRC), six historic cities have recognized assets:

- Pelotas: Sweet Traditions of Pelotas Region and Old Pelotas (Arroio do Padre, Capão do Leão, Morro Redondo, Turuçu);
- Paranaguá: Mullet fishing from Ilha do Mel;
- Angra dos Reis: Venerable Brotherhood of Saint Benedict;
- Corumbá: Bath of São João of Corumbá and Ladário (MS);
- Porto Seguro: Open Museum of the Discovery;
- Parnaíba: Santeira Art of Piauí;
- São João del Rei: The Language of Bells in Minas Gerais Artisan cheese "Minas type".

The state of Ceará is the only one to have a state-level inventory of intangible assets, with the category "Living Treasures of Culture", in which Sobral had two registered masters: Mestre Panteca, guardian of the Boi Bumbá tradition, and Mestra Rita de Cássia, guardian of the tradition of the Doceira de fartes.

The procedures involving the descriptive movement allowed us to recognize that, among the 11 target municipalities of this study, the municipalities of Pelotas/Rio Grande do Sul, São João del Rei/Minas Gerais, and Sobral/Ceará showed indications of the existence of a creative territory based on gastronomy. Only the three have institutionalized recognition that gastronomy is intangible heritage which makes up cultural identity [5,15,18]. In Sobral, it was not possible to deepen the studies because the municipality was excluded in the later stages of the method, since there was no granting of an interview by the public management.

In São João del Rei, it was identified that local gastronomy is directly associated with the process of social and economic development of the municipality [9,22], in which the socio-spatial relations around the marketing and consumption of artisanal cheese "Minas type" were established in the historic center.

Retaking the approach of Funari and Pelegrini [13], it is observed that in São João del Rei, cultural practices about the marketing and consumption of cheese are linked to the built cultural heritage, because it is in the historic center where one can find commercial houses specialized in cheese and derivatives. In parallel is an appreciation of the small producer [13] and of artisanal techniques [13,18].

Socio-spatial relations are also verified in the scope of domestic hospitality, since the act of receiving a person through the kitchen door or the living room door says about the degree of intimacy one wants to establish with the visitor. Receiving through the kitchen in São João del Rei shows the welcome the host wishes to express [16], while receiving through the living room represents formality and distance in the relationship with the visitor [7].

Regarding identity, São João del Rei has in the artisanal cheese one of the cultural elements that make up its cultural identity [5,15,18,19].

In Pelotas, the Traditional Sweets are associated, at first in the history of the city, to the development of cultural practices, since it was services in soirees that took place inside the houses, located in what is now understood as the historic center, which allowed

the construction of identity values coupling the sense of belonging [5]. Nowadays, the Traditional Sweets still play this cultural role, and, subsequently, they have also assumed a utilitarian role [7,8], which is responsible for the largest business fair in the region, which occurs annually for 19 consecutive days (Table 2).

**Table 2.** Synthesis of the Cultural Dimension of Tourism in Brazilian Historic Cities.

| Municipality | Date of Toppling | Cultural Attractions | Ongoing Events | Heritage Preserved by IPHAN | State Protected Assets | National Inventory of Cultural References (INRC)—IPHAN |
|---|---|---|---|---|---|---|
| Pelotas/RS | 2018 | 38 | 4 | 23 | 9 | 1 |
| Paranaguá/PR | 12/3/2009 | 37 | 3 | 4 | 26 | 1 |
| Mambucaba Historical Village—Angra dos Reis/RJ | 1969 | 33 | 11 | 7 | 21 | 1 |
| Cabo Frio/RJ | 1989 | 17 | | 7 | 5 | 0 |
| Nova Friburgo/RJ | 1972 | 28 | | 7 | 13 | 0 |
| Petrópolis/RJ | 1964 | 73 | 3 | 10 | 38 | 0 |
| Corumbá/MS | 9/28/1993 | 19 | 2 | 15 | 1 | 1 |
| Porto Seguro/BA | 1974 | 33 | 4 | 11 | 0 | 1 |
| Sobral/CE | 2000 | 19 | | 5 | 0 | 0 |
| Parnaíba/PI | 2011 and 2017 | 21 | 3 | 2 | 2 | 1 |
| São João del Rei | 1938 | 41 | 8 | 6 | 0 | 1 |

Source: Organized by the Authors.

*4.2. Economic Dimension*

When analyzing the information of the Economic Dimension of the Historic Cities under study, it can be seen that the two most populous in order are Pelotas and Petrópolis.

According to the REGIC [23], four municipalities were categorized as Regional Capital (Pelotas, Cabo Frio, Petrópolis, and Sobral) because they exert great regional influence, since they are educational poles and have industries and services such as hospitals, banks, commercial centers, and many other goods and services. The other seven municipalities received the Sub-Regional Center classification; this implies that they influence other smaller cities but are subordinated to other regional capitals.

The four Regional Capitals and the Sub-Regional Centers of Corumbá and São João del Rei form their own Population Arrangement. The expression Population Arrangement refers to the functional connection existing between two or more municipalities. This connection occurs from the movement of residents to work, study, or even by forming urban agglomeration in a contiguous way. The occurrence of population arrangement in São João del Rei explains why it became part of this research, since it is not a medium size city by itself, but we considered its population arrangement with Coronel Xavier Chaves, Tiradentes, and Santa Cruz de Minas. Corumbá is the only municipality that forms an international population arrangement since it is located in the borderland and has connections with cities in Paraguay and Bolivia.

Among the Regional Capitals, only Petrópolis does not have an airport. Four municipalities have international airports, and two municipalities have regional airports. Five municipalities do not have an airport within their territory, being served by airports within a radius of up to 200 km. Sobral's Regional Airport is currently in the process of being tendered.

In the classification of the Brazilian Tourism Map, carried out by the Ministry of Tourism, only the municipality of São João del Rei has its tourist region linked to its historical aspect, Trilha dos Inconfidentes. Other seven municipalities have the names

of tourist regions referring to their geographical aspects; they are Costa Doce, Litoral do Paraná, Costa Verde, Costa do Sol, Pantanal, Vale do Acaraú, and Polo Costa do Delta.

Three municipalities, which are in two tourist regions—two municipalities share the same tourist region—make a combination between historical and geographical aspects: Serra Verde Imperial and Costa do Descobrimento. In the nomenclature of tourist regionalization, none of the historic towns emphasize their local gastronomy. Montanari [16] understands that typical local gastronomy is formed by a combination of elements available in a territory, such as spices, climate, geographical aspects, local rituals, and others. Although Pelotas has a sweet tradition recognized by IPHAN, in the historic city, which belongs to the touristic region of the Sweet Coast, the word "Sweet" does not allude to its gastronomic culture; it refers to its geographical aspects, because it is a coast bathed by sweet waters.

With regard to the number of travel agencies, Porto Seguro is in first place, with 83 agencies. Soon after comes Petrópolis with 72 agencies. The city with the lowest number of agencies is Sobral, with nine agencies.

Cabo Frio has 796 lodging facilities, a much higher number when compared with the second municipality with the highest number, Porto Seguro with 479 facilities. This discrepancy between the historic cities becomes even more evident when compared to the smaller numbers, Sobral and Corumbá, each with 14 means of lodging.

Among the municipalities with information about food services, Pelotas has 253 food services, ranking first. In Angra dos Reis, the Mambucaba Village is in a geographically more distant location from the city; for this reason, only the food services installed in the Village were considered, being 10 services.

The participation of the accommodation and food sectors in the general employment of the municipalities are presented here in percentages, since it was previously demonstrated in quantitative and separated by sector. Through these percentages, we can see that the municipality of Porto Seguro has its economy based mainly on cultural and tourist activities, because 37.36% of general jobs are linked to the direct sectors of tourism, which are food and accommodation.

Even though Cabo Frio has the largest number of lodging facilities among the historic cities, its employability is in second place, with 12.90% of overall jobs associated with the accommodation and food service sectors. Sobral is the municipality with the lowest participation of the lodging and food service sectors in overall employability, with 3.01% of jobs.

About food services, the Village Mambucaba (Angra dos Reis) is located geographically far from the city center, so we considered only the food services installed in the Village, being 10 services. São João del Rei has 53 food services in its historical center, most of them offering artisan cheese or some dish that uses cheese. Pelotas has 253 food services, ranking first among the 11 cities.

The intangible gastronomic heritage in the historic cities of Pelotas and São João del Rei played different roles in its origins, the former being a cultural role and the latter an economic role. In the course of the history of both cities, the social relations that occurred in the space of the historic centers instilled in them other meanings.

In the case of the Traditional Candies of Pelotas, the economic contribution [6] glimpsed from the creation of the National Candy Fair in 1986, with annual editions, increased the demand for professionals who worked in production and commercialization and prospected the identity relevance [5] for the municipality. In São João del Rei, the cheese had an inverse social relationship than occurred with the Traditional Candies of Pelotas. At first, it emerged as an economic alternative to gold exploitation, and contemporarily it is an identity structurer, a symbol of the mining culture, which extrapolates the territory of the municipality [5,15,16,18]. The artisanal cheese from São João del Rei is sold throughout Brazil.

In Table 3, a summary of the Economic Dimension of Tourism in the Brazilian Historic Cities studied here is presented.

**Table 3.** Summary of the Economic Dimension of Tourism in Brazilian Historic Cities.

| Municipality | Inhabitants (Estimate 2020) | MAPA Classification | Travel Agency | Means of Accommodation | Food Services | Participation of the Accommodation and Food Sectors in Overall Employerability |
|---|---|---|---|---|---|---|
| Pelotas/RS | 343.132 | B—Costa Doce | 24 | 44 | 253 | 5.53% |
| Paranaguá/PR | 156.174 | B—Litoral do Paraná | 24 | 81 (18 Sede e 63 Ilha do Mel) | 210 | 4.75% |
| Mambucaba Historical Village—Angra dos Reis/RJ | 207.044 | A—Costa Verde | 46 | 345 | 10 (only in Village) | 9.20% |
| Cabo Frio/RJ | 230.378 | A—Costa do Sol | 27 | 796 | not available | 12.90% |
| Nova Friburgo/RJ | 191.158 | B—Serra Verde Imperial | 26 | 115 | not available | 4.52% |
| Petrópolis/RJ | 306.678 | A—Serra Verde Imperial | 72 | 181 | 155 | 7.90% |
| Corumbá/MS | 112.058 | B—Pantanal | 36 | 14 | 145 | 4.87% |
| Porto Seguro/BA | 150.658 | A—Costa do Descobrimento | 83 | 479 | not available | 37.36% |
| Sobral/CE | 210.711 | B—Vale do Acaraú | 9 | 14 | not available | 3.01% |
| Parnaíba/PI | 153.482 | B—Polo Costa do Delta | 37 | 33 | 197 | 7.17% |
| São João del Rei | 90.497 | B—Trilha dos Inconfidentes | 18 | 47 | 53 | 6.53% |

Source: Organized by the Authors.

### 4.3. Political Dimension

In the synthesis of the Political Dimension of the historic towns, it can be seen that all the historic towns have Municipal Tourism Councils and Municipal Councils of Culture, thus demonstrating that structurally all of them are politically adequate. Most of the Cultural Councils are active and have a Municipal Plan of Culture. On the other hand, when we analyze the role of the Tourism Councils, we notice that many seem to exist only in formalization, since most municipalities do not have a Municipal Tourism Plan. Four municipalities have their Municipal Tourism Plans available for broad access; they are Pelotas, Paranaguá, Petrópolis, and Corumbá.

As for the existence of a Convention and Visitors Bureau, two historic cities do not have a CVB: Corumbá and Sobral. Six municipalities have their own CVB: Paranaguá, Angra dos Reis, Cabo Frio, Nova Friburgo, Petrópolis, and Porto Seguro. Parnaíba and São João del Rei have regional CVBs because they belong to tourist regions in which they are located. The city of Pelotas has already had a CVB, but it is currently deactivated, which shows a certain disarticulation of the tourist and cultural trade of the municipality. The presence of a CVB contributes significantly to the development of tourist and cultural activities, for it participates actively in the formation of public policies, in the training of professionals, and in the promotion of the destination.

In the political structuring of public management, it is observed that the area of culture, again, seems to be better articulated. With the exception of Sobral, which has only one coordinator of culture, the other municipalities have a political administrative division with the word "culture" in its title, which points to the political strength of the cultural segment. Tourism appears in a more diffuse form in the political structure, since most of the historical cities have agglutinated the area with other sectors, such as innovation, leisure, marketing, and sports, possibly pointing to a development of actions directed to the economic sector, both of culture and of tourism. The municipality of Sobral does not

contemplate the tourism sector in its political organization. Angra dos Reis also does not have a department or sub-department that contemplates tourism. Paranaguá, Porto Seguro, and São João del Rei have a Secretariat of Culture and Tourism, perhaps indicating the strengthening of public policies focused on historical and cultural aspects.

In the Political Dimension, only Pelotas demonstrates to structure the official cultural identity of the municipality in the relationship between built cultural heritage and gastronomy. In 2012, the Association of Candy Producers of Pelotas–RS received the Geographical Indication (GI) in the modality Indication of Source [24]. Pelotas has turned its attention to the development of creative activities that link the built cultural heritage and the gastronomic intangible heritage [9,22]. To this end, three institutional documents were prepared: the Municipal Cultural Plan for the first decade 2022–2032; the Municipal Tourism Plan of Pelotas–RS for 2020–2024; and the Positioning and Strategies for Tourism in the Rio Grande do Sul Sweet Coast. This last document has a regional scope, and in it the city of Pelotas is treated as "Creative Pelotas", due to the projects and actions developed to stimulate and strengthen the creative territory, especially gastronomy.

In São João del Rei, despite its strong identity relationship with artisanal cheese, what was identified with potentiality for the development of creative activities are those linked to the intangible heritage of music, associated with the Language of Bells [9,11]. In the other historic cities, no official documents were identified that regulate, institutionalize, or encourage the creation of a creative territory that is based on the relationship between built heritage and gastronomy.

Table 4 presents a synthesis of the Political Dimension of Tourism in the Brazilian Historic Cities studied here.

**Table 4.** Summary of the Politic Dimension of Tourism in Brazilian Historic Cities.

| Municipality | Municipal Plan of Culture | Municipal Tourism Plan | Convention Visitors Bureau | Secretariat of Culture | Secretariat of Tourism |
|---|---|---|---|---|---|
| Pelotas/RS | ✓ | ✓ | Inactive | Municipal Secretariat of Culture—SECULT | Secretariat of Development, Tourism and Innovation—SDETI |
| Paranaguá/PR | ✓ | ✓ | Paranaguá Convention and Visitors Bureau | Municipal Secretariat of Culture and Tourism | |
| Mambu-caba Historical Village—Angra dos Reis/RJ | - | - | Angra e Ilha Grande Convention and Visitors Bureau | Secretariat of Culture and Heritage | - |
| Cabo Frio/RJ | ✓ | - | Cabo Frio Convention and Visitors Bureau—CFCVB | Secretariat of Culture | Secretariat of Tourism, Sport and Leisure |
| Nova Friburgo/RJ | ✓ | - | Nova Friburgo Convention and Visitors Bureau | Municipal Secretariat of Culture | Municipal Secretariat of Tourism and City Marketing |
| Petrópolis/RJ | ✓ | ✓ | Petrópolis Convention and Visitors Bureau | Municipal Institute of Culture and Sports | Turispetro Secretariat |
| Corumbá/MS | ✓ | ✓ | - | Foundation of Culture and Historical Heritage of Corumbá | Pantanal Tourism Foundation |
| Porto Seguro/BA | ✓ | - | Porto Seguro Convention and Visitors Bureau | Secretariat of Culture and Tourism | |
| Sobral/CE | ✓ | - | - | Secretariat of Youth, Sports and Leisure | |
| Parnaíba/PI | ✓ | - | Litoral Piauiense Convention and Visitors Bureau | Superintendent of Culture | Superintendency of Tourism |
| São João del Rei | ✓ | - | Trilha dos Inconfidentes Convention and Visitors Bureau | Secretariat of Culture and Tourism | |

Source: Organized by the Authors.

## 5. Conclusions

Following the presentation of the results by dimensions, we will also present the theoretical and practical implications by the dimensions assumed in this dimension in relation to the creative tourism territory. In the Cultural Dimension, three cities stand out for having a larger quantity of activities related to culture: Petrópolis (73 attractions), São

João del Rei (41 attractions), and Pelotas (38 attractions); five have centrality of visitors displacement for cultural purposes: Pelotas, Cabo Frio, Porto Seguro, Parnaíba, and São João del Rei. Only Pelotas, Sobral, and São João del Rei have gastronomic intangible heritage, the first two linked to regional sweets and the third focused on cheese production derived from dairy production. The three historic cities are regional references in the relationship between tourism and gastronomy.

In the Economic Dimension, four cities were categorized as Regional Capital (Pelotas, Cabo Frio, Petrópolis and Sobral). Porto Seguro is doubly outstanding for having the largest number of travel agencies (83 agencies) and having 37.36% of the general jobs linked to the direct tourism sectors, which are food and lodging. Pelotas has the largest number of food services, with 253 services. Cabo Frio has the largest number of lodging facilities, with 796.

In the political dimension, it is observed that the cultural sector seems to be better organized than the tourism sector. Both sectors in all the historic cities have Tourism and Culture Councils, but when it comes to Municipal Plans, all the municipalities have a Culture Plan, but only four have a Tourism Plan: Pelotas, Paranaguá, Petrópolis, and Corumbá. As for the existence of a Convention and Visitors Bureau (CVB), three historic cities do not have a CVB: Pelotas, Corumbá, and Sobral.

Resuming the objective of this study, which was to characterize the tourism of the Brazilian historic cities, it is verified that in the Cultural Dimension all of them have a diversified range of cultural attractions directed to their built cultural patrimony. However, only three, Pelotas, Sobral, and São João del Rei, have gastronomy as one of the cultural aspects that make up their cultural identity. São João del Rei, despite understanding cheese as a structuring part of its identity, does not promote actions to promote gastronomic cultural heritage; thus, only Pelotas and Sobral have social and economic relations that configure a creative territory which is based on gastronomy.

In the Economic Dimension, it can be observed that only Porto Seguro has its economy based on the tourist activity, but the tourist activities are mostly focused on nature, which propitiates the development of sun and beach tourism, relegating cultural and creative tourism to a second plan. Cabo Frio and Angra dos Reis also have a significant portion of their economy focused on tourism, and just like Porto Seguro, they are dedicated to sun and beach tourism.

Finally, the Political Dimension demonstrates that tourism in the historic cities still presents several fragilities in what concerns heritage valuation, both material and immaterial. Pelotas seems to be the historic city that emerges the most as a cultural and creative tourism destination focused on its intangible cultural heritage with the Traditional Sweets. The gastronomical heritage is understood as an important experience [19]. However, it still does not play a prominent role when compared to the official identity of the historic cities.

Thus, the Political Dimension shows that the cultural heritage that is valued is still the one linked to the economic elites, that is, the built cultural heritage. Gastronomy, an intangible cultural heritage, which should value the peripheral world [13], is politically forgotten in the historic towns.

Henri Lefebvre's regressive-progressive method allowed a dialectical analysis of the historic cities through cultural tourism, which made it possible to study the general processes, including continuities and discontinuities, merging the particularities to the national context. Identifying the historic towns as mediators of their own articulations between the general and the particular places in evidence, by means of the cultural, economic, and political dimensions, the contradictory and conflicting movement experienced in the course of their histories, however, without presenting solutions to dissolve them.

The theoretical and empirical contributions emerge from the results that demonstrated that the creative territory, based on gastronomy, in the historic cities is not yet a reality in Brazilian cities—Pelotas was the exception. In this sense, the contribution of this study deals with the little cultural appreciation of the historic centers of the historic towns and the lack of stimulus on the part of local administrators for the creation of creative

territories based on local gastronomy. The existing tourism in these historic towns is essentially linked to the existence of the built heritage, and gastronomy is offered as a form of biological restoration and not as a local cultural experience. The cultural identity in these cities has been presented based on the architectural heritage of the European colonizers, disregarding their food heritage and that of any other ethnic group present there. The collective memory of food practices is restricted to the domestic sphere of local residents. The territory of the historic centers of the historic towns is permeated by economic, cultural, and political power relations, being perceptible both in their spatial distribution and in the social relations that take place in the historic center. The use of this epistemological and methodological approach has proven to be very effective for tourism studies, especially for those who seek in the cultural root the compression of the phenomenon, therefore being a scientific-methodological contribution of this research.

The declaration by the United Nations on 11 March 2020, that the world was experiencing a new pandemic of the COVID-19 virus, transmissible by the respiratory tract, significantly compromised the data collection stage. It prevented going to the field because tourism and cultural activities were paralyzed in 2020 and 2021.

Finally, it is suggested that new research be conducted with the employment of the epistemological approach of the regressive-progressive method in Brazilian historic cities from the perspective of cultural tourism, analyzing socio-spatial relations in creative territories based on other intangible heritage. It is also worth suggesting the replication of the methodological procedures employed in the historic towns for other cities and other cultural contexts, relating cultural tourism and creative territories.

**Author Contributions:** Supervision, F.A.d.A.; methodology and language correction, A.L., F.A.d.A. and R.C.; writing and editing, A.L.; analysis and data correction, R.C. All authors have read and agreed to the published version of the manuscript.

**Funding:** Coordenação de Aperfeiçoamento de Pessoal de Nível Superior (CAPES)—Brazil—Financing Code 001.

**Institutional Review Board Statement:** Not applicable.

**Informed Consent Statement:** Not applicable.

**Data Availability Statement:** All data included in this study are available upon request by contacting the corresponding author.

**Acknowledgments:** We are grateful to anonymous reviewers for their constructive reviews on the manuscript and the editors for carefully revising the manuscript.

**Conflicts of Interest:** The authors declare that they have no conflict of interest.

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
