# Peer review of "Creative Territory and Gastronomy: Cultural, Economic, and Political Dimensions of Tourism in Historic Brazilian Cities"

_sustainability, doi:10.3390/su15075844_

Round 1

Reviewer 1 Report (Previous Reviewer 4)

I think that the authors have significantly improved their article. I find this article interesting and worth being published with minor revisions. I would suggest making the following improvements:

1 - The abstract is too long. I think it could be shortened.

2 - The text of the article could be better structured. Please mind the size of the paragraphs and the coherence of the narrative line.

3 - Line 479 should be translated into English.

4 - The sources of the tables should be better identified. How, when, and where did you collect the data? Ex.: Authors' table based on data from ....

5 - I don't understand the last two paragraphs in the Conclusion. Lines 541-542: If you say "First", logically the reader waits for "Second" and/or "Third". Lines 543-548: I don't understand what you want to say here. Could you maybe rephrase this paragraph?

Author Response

Reviewer 2 Report (New Reviewer)

The proposed research is interesting and may inspire other researchers. However, it raises a few of my remarks:

1. The term ‘creativity’ has a very extensive literature that has not been discussed here. I see that the authors should refer to the rich literature on creative cities as well as creative territories.

2. The authors use the term ‘creative territories’, which is neither theoretically nor operationally discussed, which reduces the methodological value of the term introduced.

3. I believe that the numerous introduced terms, such as cultural heritage, gastronomy, tourism, food heritage, and many others, should be defined for the purposes of research, and a model of hypothetical relationships between all discussed social phenomena should be presented, defined in a precise way. Such a methodological approach will build a platform for continuing research, e.g. in other countries.

In conclusion, I believe that the article contains methodological shortcomings that will explain the results obtained. The authors had adopted certain theses, which are expressed implicitly and not explicitly.

After completing the definitions and hypothetically assumed relationship between described phenomena, the article will be an interesting study that may be an inspiration for research in other countries.

Author Response

Reviewer 3 Report (New Reviewer)

Dear authors,

Thank you for accepting the suggestions. I find present form of the manuscript suitable for publication. 

Good luck with further research.

Author Response

Reviewer 4 Report (New Reviewer)

Dear Authors,

Congratulations on your interesting article. I have some suggestions that I hope will improve the general readability of your paper.

I suggest to improve your introduction by providing some background on what prompted your research - why do you consider it was necessary, how it relates to previous research, how does your research fit within the existing body of knowledge, and how does it contribute to the existing knowledge. I also suggest you give an overview of the structure of the paper.

Secondly, I suggest expanding your literature review or if you feel that there is a gap in the literature regarding your research topic, I suggest you provide details in this sense. 

Regarding your conclusions, I suggest you expand them (or include a separate discussion section) to draw attention to your findings, how they fit in within the literature, how they improve the existing knowledge and in which way they may be useful for the tourism industry.

Good luck with your research!

Round 2

Reviewer 4 Report (New Reviewer)

Dear Authors,

I went over the improvements you made to your article, now I think it meets the criteria for publication

Good luck with your future research!

This manuscript is a resubmission of an earlier submission. The following is a list of the peer review reports and author responses from that submission.

Round 1

Reviewer 1 Report

Dear Authors, I am extremely sorry to convey that the submitted manuscript in its present form is not publishable. It requires a complete overhaul and probably a resubmission elsewhere. 

Reviewer 2 Report

The paper was too descriptive and lacks academic depth. The introduction did not clearly contextualize the aim of the paper, nor the related set of objectives to meet or address the aim - so the full intent of the paper was not really clear, especially where the framing of the research problem was inadequate.

The literature review section was rather disjointed. It was constructed in a rather fragmented manner and therefore a clear conceptual framework was not apparent - the review should illustrate competence in arriving at an appropriate research framework. It would have been purposeful to have contextualized the research gaps more or justify more the need for such a 'critical enquiry'.

As the foundations were not strong then the methodological grounding was not aligned to the study too, and certainly was not convincing or progressive in any way of form. The work needed far more epistemological-based justification. 

Consequently, the finings were loose and did not create effective linking of the discussion to the research problem; nor did they adequately address the conceptual and theoretical framework (and especially - as mentioned - the conceptual framework was vague). 

The cultural, political and economic dimensions did not link to a specific sustainability framework - which would have had scholastic relevancy for the journal. 

Reviewer 3 Report

The subject of the manuscript is interesting, but I have many reservations: Main remarks

1. Introduction - The introduction is very short. It does not show the essence of the manuscript. It does not show what the research gap is.

2. It is not clear from the literature review what the approach to the Authors' review is. What are the authors' considerations? What theory do they agree on?
3. Discussion -
there is no in-depth discussion

4. Conslusion - very short. what are the directions for the future? What were the limitations?

5. References - very short. Only 14 positions

Reviewer 4 Report

I think that the topic of the research project is very ambitious and promising. However, I would suggest making some revisions to the text:

1 - I think that the general tone of the article is a bit too general. The authors put forward some interesting arguments without going too much into detail. That's why I have an impression that this paper fits more with the scientific magazine-style format rather than the academic one. In particular, I would be interested in seeing a more detailed presentation of the research findings (section 4 "results"). It might be useful to divide this section in several subsections (as per cultural, economic, political dimensions) and discussing each issue more thoroughly.

2 - In the title of the paper you say that you are going to study the dynamics of socio-spatial relations. This means that your research should be framed within a particular time framework. I have found it a bit difficult to follow a clearly defined historical path in your research project. I would be interested in seeing the concrete evolution of socio-spatial relations in the geographical region you touch upon in your paper, so that the reader could understand how the situation has changed over time. 

3 - I think you could add more illustrations to the paper, notably maps. The term "socio-spatial relations" presupposes that your paper shall include at least several maps drawn with professional software (Khartis for example).

4 - The Introduction of your paper is too brief, in my opinion. For the non-Brazilian audience it might be hard to get the idea right away. I would suggest adding 2-3 paragraphs and explain more in detail the context, the rationale, and the objectives of your study.

5 - Lines 123-129 in Literature Overview are not well referenced.

6 - I think that the overall focus of this paper is more biased towards "creative territory" rather than "gastronomy". I would be interested to learn more about the impact of your study on Brazil's gastronomy industry, apart from the cultural heritage argument.

Reviewer 5 Report

First of all, we would like to congratulate the authors for their work and dedication in carrying out this study. The subject of the article sounds like a very interesting one. 

The identified shortcomings in the article:

I want to mention that the suggestions represent only the personal opinions of a simple reader of the article, aiming at the improvement and a better understanding of the manuscript.

From my point of view, the article doesn`t have the potential to be accepted unless major interventions are made

The title itself is a bit forced since gastronomy is almost inexistent or of marginal concern in almost all the analyzed Historic City, as it is presented in Results and Conclusions

Why did you choose this topic, since there is no clear connection between your data, your analysis and results?  

The abstract seems to me to be quite short and unobtrusive. It should be extended

The article is precise and clear but it does not connect the title and the reality.

The hypotheses and conclusions are clear and simple but TOO SIMPLE.

Indeed, as stated by the authors, the paper deals with the results obtained in the first of three movements - the descriptive process. We acknowledged that, but the article sounds from the beginning to the end, too descriptive, too linear, without any intrigue addressed.

The article is not very well documented, has only a few bibliographic sources, some of them of great importance and relevance for the subject in question. The literature review is also very short and based on translated editions or local impact scientific work

The entire scientific argument is overshadowed by the austere bibliography and the lack of a direct link between the title and the results obtained

The only method that stands out after reading the article is the descriptive one, with a poor sense of correlation

Seems like some analyzed data are also taken from descriptive documents – see line 212 to 213 – All the HC studied have …. events dedicated exclusively to artisanship. Where is that information? Is it important for your dissertation?

The map is of poor quality and the information presented is very low – I suggest changing the map (as it looks right now, it cannot be considered a map)

At the same time that globalization promotes the erasing of territorial singularities, 123 it also stimulates, paradoxically, the marking and reinforcement of regional and national 124 identities as a way to differentiate from other territories, from other cultural identities. 125 This paradoxical tendency contributes to the consolidation of national identities and re- 126 gional units. 127 … I really doubt this is author’s ideas and there is a need of citation

These are just part of the weaknesses identified - I strongly suggest to authors to pay more attention and literature review and also to try to avoid description  when it comes to deliver different categories of data